# Spatial Uncertainty in Wildfire Forecasting Using Multi-Modal Earth Observation

## Abstract

Accurate wildfire forecasting from remote sensing data is essential for climate resilience and emergency planning. Beyond predictive performance, understanding where and why uncertainty arises is critical for operational trust. We analyze the spatial structure of predictive uncertainty in wildfire spread forecasts using multimodal Earth observation (EO) inputs, including Sentinel-2 vegetation indices and VIIRS thermal reflectance. Using Monte Carlo dropout, Deep Ensembles, and Bayesian Neural Networks for uncertainty quantification, we find that uncertainty estimates are spatially structured and concentrated near predicted fire perimeters, consistent with the expected uncertainty in fire spread forecasts. We introduce a novel and interpretable centroid-oriented distance metric that reveals high-uncertainty regions consistently form 20–60 meter buffer zones around predicted firelines. Feature attribution using integrated gradients highlights vegetation condition and recent fire activity as primary drivers of model confidence. Together, these results suggest that spatial uncertainty in EO-based wildfire forecasting is structured, interpretable, and operationally actionable. The code for all experiments is available on GitHub.[1]

## 1 Introduction

Wildfires have become an escalating global crisis, intensified by climate change, prolonged droughts, and expanding human development. Most recently in January 2025, Southern California experienced wildfire events that have been among the costliest natural disasters in U.S. history. Wildfires in the European Union have become increasingly frequent and severe, with over 166,000 hectares burned by May 2025, nearly three times the long-term average, driven by climate change and affecting regions beyond the traditional Mediterranean hotspots (EFFIS, 2025). Globally, regions such as the Amazon, North America, Australia, and parts of Africa have witnessed unprecedented wildfire activity, leading to significant ecological damage, loss of biodiversity, and adverse health effects due to smoke exposure (Cunningham et al., 2024). As climate change continues to exacerbate wildfire risks, there is an urgent need for accurate, high-resolution wildfire forecasting to aid in early response, resource allocation, and risk mitigation. While remote sensing products like VIIRS (Schroeder et al., 2014) and MODIS provide near real-time fire detections, they do not forecast how wildfires will evolve in the days to come.

Traditionally, fire spread forecasting has relied on physics-based simulators such as Farsite (Finney, 1998) and Prometheus (Tymstra et al., 2010), which use hand-crafted rules and environmental inputs to simulate fire growth. These tools are interpretable and physically grounded but require fine-grained inputs—like fuel maps and localized weather forecasts—which are difficult to obtain in real time and hard to calibrate to dynamic fire conditions.

Machine learning has emerged as a scalable alternative for fire forecasting (Radke et al., 2019; Bolt et al., 2022), learning directly from remote sensing and historical fire data. Shadrin et al. (2024) trained U-Net and DeepLabV3 models to perform multi-day fire spread segmentation using multimodal remote sensing and meteorological inputs. Their work benchmarks a variety of input combinations and shows strong predictive

---

[1] https://github.com/roloccark/wildf-UQ

performance, but does not quantify uncertainty—leaving users without insight into where or why the model might be wrong.

In parallel Huot et al. (2022) introduced the NextDayWildfireSpread dataset, emphasizing single-frame predictions of fire growth based on remote sensing. Gerard et al. (2023) extended this effort by releasing WildfireSpreadTS, a large-scale benchmark for multi-temporal wildfire forecasting across 607 fire events. The dataset incorporates Sentinel-2, VIIRS thermal bands, fire history, meteorology, and slope, enabling testing of temporal models like ConvLSTM Shi et al. (2015b) and U-Net with Temporal Attention Encoder (UTAE (Garnot & Landrieu, 2021)).

Generative modeling is also being explored for fire forecasting. Shaddy et al. (2024) use a physics-informed GAN (cWGAN) to fuse fire simulations from WRF-SFIRE with satellite observations. Their system outputs ensembles of arrival time maps conditioned on sparse input detections, providing uncertainty-aware predictions that can be used to initialize atmospheric-fire models. However, their approach focuses more on generating plausible initial states than on operational fire masks or end-to-end spatial uncertainty analysis using public datasets.

Despite the operational risks involved, no prior work has investigated uncertainty quantification in high-resolution wildfire forecasting. Most existing approaches are fully deterministic, producing binary or probabilistic predictions without expressing model confidence—leaving critical questions of when and where the model may fail unanswered. This omission is especially concerning for wildfire response, where uncertainty-aware decision-making is essential for frontline planning, containment strategies, and risk assessment. In this paper, we take a first step toward addressing this gap by analyzing the spatial structure of predictive uncertainty in Earth observation-based wildfire forecasts. Using Monte Carlo dropout and attribution techniques, we investigate where uncertainty arises, how it aligns with vegetation and fire morphology, and how it could guide the construction of interpretable buffer zones to support triage and operational planning.

## 2   Why This Study

Our work focuses on the operational quantification of uncertainty in deep learning-based, pixel-wise wildfire spread prediction. While prior efforts have tackled deterministic segmentation, event-level uncertainty, probabilistic fire danger indices, surrogate-assisted physical modeling, and generative reconstruction of fire histories, these approaches often lack the spatial resolution or reliability needed for on-the-ground decision-making. We aim to directly benchmark and interpret spatial uncertainty in multi-day, high-resolution spread forecasts using real-world wildfire events. By emphasizing calibrated and interpretable uncertainty at the pixel level, our work bridges the gap between state-of-the-art machine learning and the practical needs of fire managers and scientists operating in high-risk environments.

## 3   Methods

**Model**   We use the UTAE model (Garnot & Landrieu, 2021), a transformer-based spatiotemporal encoder-decoder architecture designed for multitemporal satellite image time series. UTAE has previously shown strong performance on change detection and land cover segmentation tasks using Sentinel-2 data, and is well-suited for the wildfire spread forecasting setting, where temporal patterns are key. Although newer transformer-based architectures, such as Swin Transformers (Liu et al., 2021), have been explored for spatiotemporal wildfire modeling (Lahrichi et al., 2025), using the same dataset, the evidence shows that they do not outperform UTAE in next-day wildfire spread prediction. On the downside, these larger architectures contain roughly 27M parameters compared to UTAE's lightweight 1M parameters, requiring significantly more computational resources for both pretraining and fine-tuning. They are also prone to overfitting and typically demand much larger datasets to generalize effectively—an unrealistic requirement given the size and variability of current wildfire datasets. For these reasons, and given UTAE's proven reliability and favorable trade-off between performance, computational efficiency, and robustness, we adopt UTAE as our primary model. As a baseline comparison, we also experiment with a ConvLSTM model (Shi et al., 2015a), a recurrent architecture tailored for spatiotemporal prediction. ConvLSTM has the ability to capture local

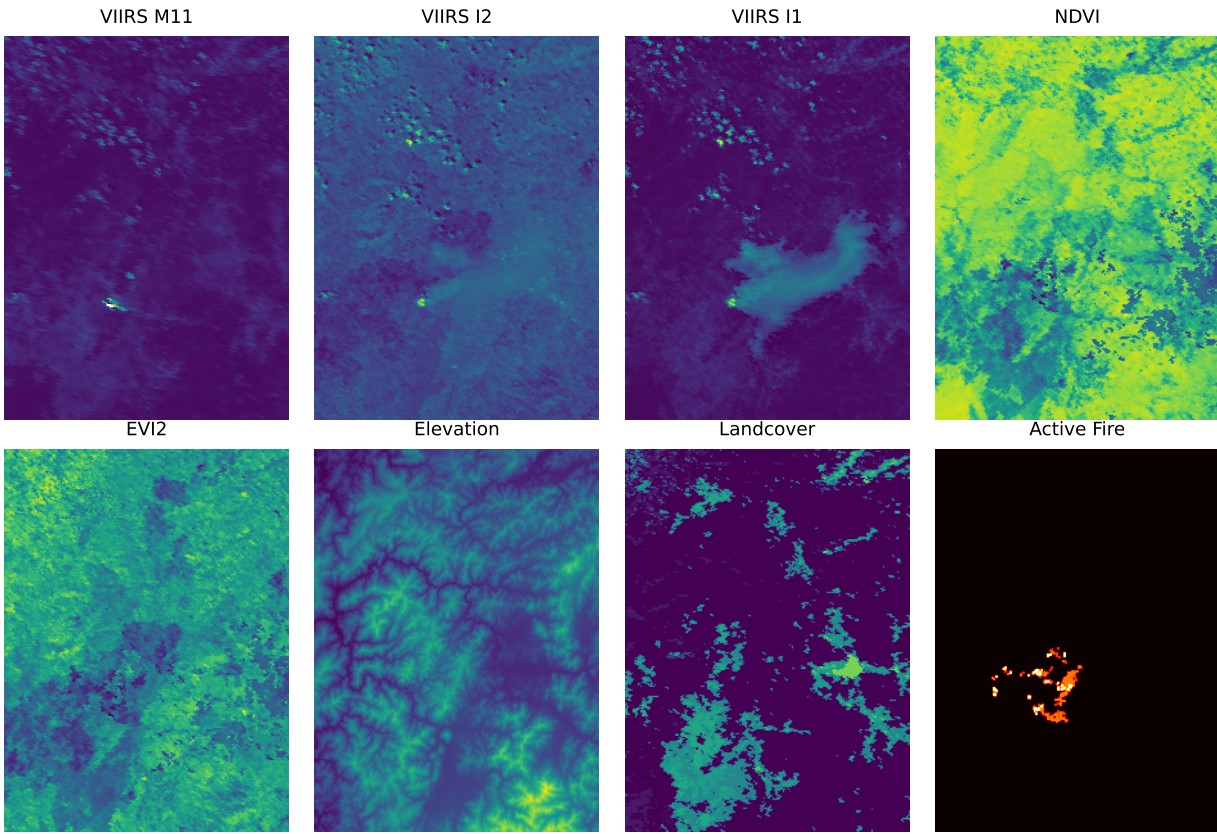

Figure 1: Example input channels from a single sample at prediction time, including Sentinel-2 bands, NDVI, EVI2, and active fire features. These inputs are provided as a 5-day sequence to the model.

temporal dependencies using convolutional operations, and has been used in wildfire modeling in prior work (Burge et al., 2022).

**Dataset** We conduct all experiments on the publicly available WildfireSpreadTS dataset[2] (Gerard et al., 2023), which provides spatial-temporal cubes of 64×64 patches centered on active wildfire regions. Each sample consists of 5 days of multimodal input features (Sentinel-2 reflectance bands, meteorological variables, NDVI, slope, and other static features), and a binary burn mask for a future day as target. The dataset includes fires from 2018 to 2021 across diverse regions. Following the original benchmark protocol, we perform 12-fold cross-validation over all year-based train/val/test permutations to account for inter-annual variability and covariate shift. A sample of the input modalities used by the model—including reflectance bands, vegetation indices, and active fire features—is visualized in Figure 1. The dataset includes 607 wildfire events across the western United States from January 2018 to October 2021, with a total of 13,607 daily images. These fires span diverse ecosystems and terrain across states like California, Oregon, and Washington.

**Training, Evaluation and Uncertainty Quantification** The model is trained in a sliding-window fashion. For each fire event, we extract 5-day sequences as input and predict the binary burn mask on the 6th day. Additional details are provided in the appendix. Spatial crops (128×128) are used to batch variable-sized regions. Following the original benchmark configuration, we train our model using only five vegetation-based input channels: VIIRS bands M11, I2, I1, NDVI, and EVI2. No meteorological or static

---

[2]https://github.com/SebastianGer/WildfireSpreadTS

terrain features are used. We report Average Precision (AP) on test folds, along with probabilistic calibration metrics described below.

We evaluate three uncertainty quantification (UQ) approaches for pixel-wise wildfire spread prediction:

1. **Monte Carlo (MC) Dropout** (Gal & Ghahramani, 2016): Dropout layers remain active at test time, and 20 stochastic forward passes are performed. The per-pixel mean and variance of predicted probabilities are used to quantify epistemic uncertainty.

2. **Deep Ensembles**: Multiple independent UTAE models are trained with distinct random seeds, each also employing MC Dropout with 20 stochastic forward passes at inference. Predictions from these models are aggregated to quantify uncertainty stemming from variations in weight initialization, training stochasticity, and dropout-induced randomness. This approach typically improves calibration and robustness.

3. **Bayesian Neural Networks (BNN)**: BNNs treat the weights of the neural network as random variables with learned probability distributions rather than fixed values (Blundell et al., 2015). We adopt a variational approximation (Bayes-by-Backprop), where each weight has a mean and variance parameter learned during training. At inference, multiple stochastic samples of the weight distributions are drawn, and predictive probabilities are averaged per pixel to obtain both a mean wildfire burn probability and an associated epistemic uncertainty map. This provides a theoretically grounded Bayesian treatment of uncertainty for pixel-wise classification, albeit with a far greater computational cost and more challenging setup and optimization compared to MC Dropout or Deep Ensembles.

To assess probabilistic quality, we compute three standard metrics: (1) Negative Log-Likelihood (NLL), which penalizes overconfident mispredictions; (2) Brier Score (Brier, 1950), a proper scoring rule for binary probabilistic forecasts; and (3) Expected Calibration Error (ECE, (Guo et al., 2017)), which measures the discrepancy between predicted confidence and empirical accuracy. All UQ experiments are implemented in PyTorch, and the code is publicly available on Github[3].

## 4 Results

Given that the model is driven purely by vegetation-based features, we interpret high uncertainty as a reflection of ambiguous fuel signatures — e.g., sparse or transitional NDVI zones, or conflicting spectral responses.

### 4.1 Feature Group Ablations

To better understand the contribution of different input feature categories, we conduct a systematic ablation study by training UTAE models using isolated subsets of inputs. Each configuration retains only one feature group—vegetation, weather, land cover, or topography—while preserving active fire indicators to ensure temporal grounding. We also include two important reference points: (i) a *persistence baseline*, which assumes that the fire remains unchanged between consecutive days, providing a non-learned lower bound on predictive skill, and (ii) an *all-features variant*, where vegetation, weather, land cover, and topography inputs are jointly supplied alongside active fire information.

Table 1 summarizes the 12-fold average precision for each setup. The persistence baseline achieves the lowest AP of $0.191 \pm 0.063$, confirming that naive extrapolation underestimates the complex dynamics of wildfire spread. Interestingly, the all-features configuration ($0.319 \pm 0.077$) does not surpass the best single-feature setup, suggesting that additional modalities introduce optimization challenges or noise under the current training regime. Among all tested UTAE variants, vegetation combined with active fire achieves the highest average precision ($0.378 \pm 0.083$), outperforming both other ablations and the ConvLSTM baseline ($0.304 \pm 0.093$). Accordingly, all downstream uncertainty analyses in this study—including spatial structure,

---

[3]https://github.com/roloccark/wildf-UQ

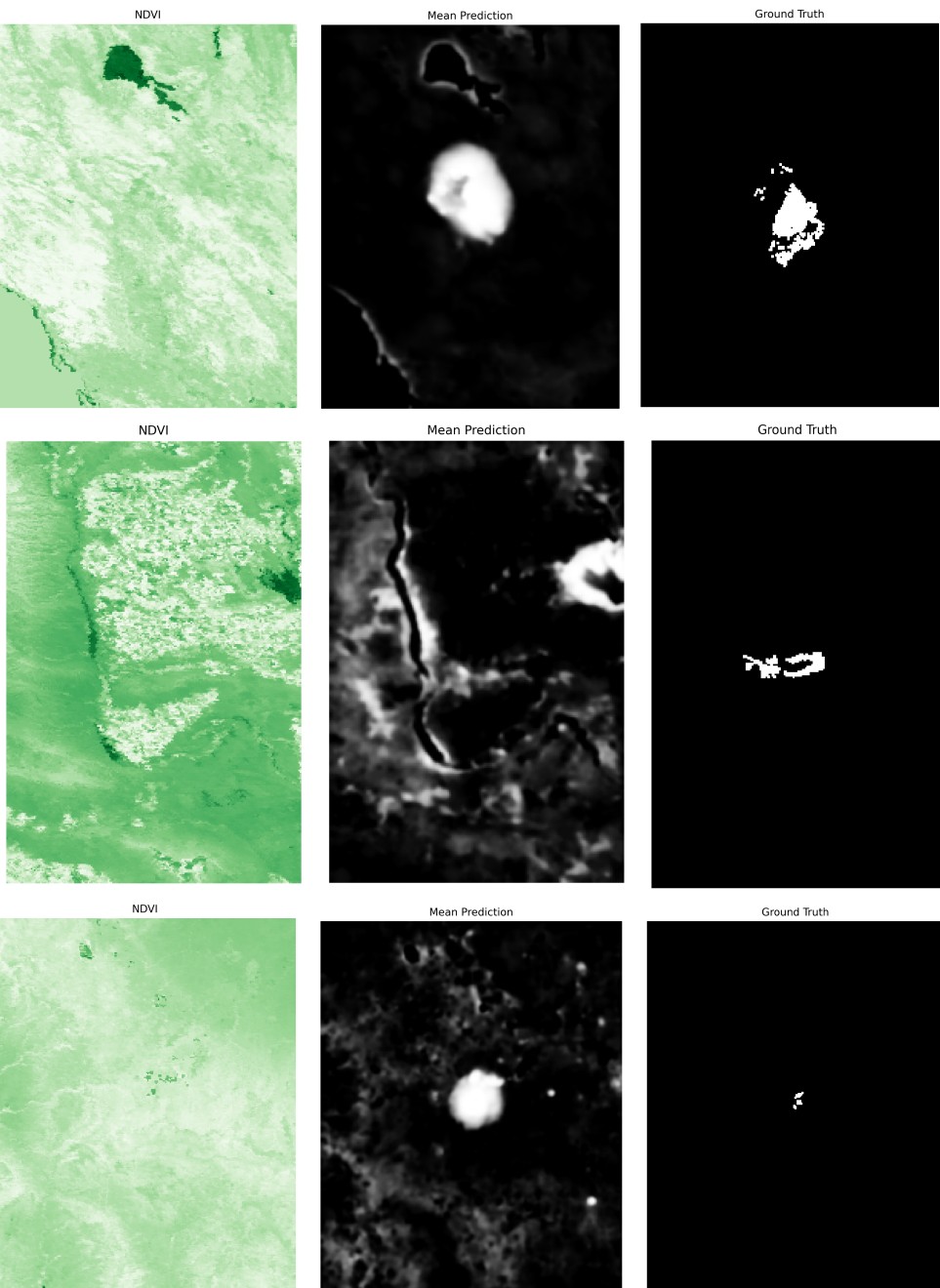

Figure 2: Qualitative comparison of model predictions for three fire events of varying size: large (top), medium (middle), and small (bottom). Each row shows the NDVI input, mean prediction from a Deep Ensemble (fold 1: trained on 2018–2019, validated on 2020), and the ground-truth burn mask. The influence of vegetation features on the model's mean predictions is clearly evident, with predicted burn areas modulated by spatial vegetation patterns. The events span approximately 125.6 acres (large), 52.3 acres (medium), and 5.2 acres (small), corresponding to 1271, 527, and 53 burned pixels respectively.

buffer zone characterization, and feature attribution—are conducted using the UTAE model trained on vegetation inputs and active fire.

Table 1: Mean Average Precision (AP) across 12 folds for different feature groups using UTAE. For comparison, ConvLSTM trained on vegetation + active fire is also included.

| Feature Group | Mean AP |
|---|---|
| Persistence baseline | $0.191 \pm 0.063$ |
| Vegetation + active fire | $0.378 \pm 0.083$ |
| Weather + active fire | $0.323 \pm 0.078$ |
| Land cover + active fire | $0.319 \pm 0.092$ |
| Topography + active fire | $0.317 \pm 0.082$ |
| All Features (veg + Weather + Land + Topo) + active fire | $0.319 \pm 0.077$ |
| ConvLSTM (veg. + active fire) | $0.304 \pm 0.093$ |

## 4.2 Benchmarking Calibration for Trustworthy Spatial Uncertainty

To ensure that our uncertainty estimates are trustworthy, we evaluate calibration using three standard metrics: Expected Calibration Error (ECE), Brier Score, and Negative Log-Likelihood (NLL). Our primary analyses—such as spatial uncertainty overlays, buffer zone estimation, and feature attribution—are based on Monte Carlo (MC) Dropout (Gal & Ghahramani, 2016) with 20 stochastic forward passes. We experimented with 10, 15, 20, 25, and 30 passes, observing that calibration performance plateaued beyond 20–30 passes. Therefore, we adopt 20 passes for computational efficiency without sacrificing reliability. To benchmark its calibration robustness, we also train a Deep Ensemble of 5 independently initialized models for each of the 12 folds. Each member of deep ensemble uses 20 MC dropout passes. The details of the BNN architecture and training procedure are summarized in Table 4. As shown in Table 2, the ensemble achieves consistently stronger calibration across all metrics compared to the MC Dropout baseline. This comparison strengthens our confidence that the uncertainty maps derived from MC Dropout are probabilistically reliable, providing a sound foundation for the spatial analyses that follow.

Table 2: Calibration metrics (12-fold averages) for the three UQ approaches. Lower values indicate better calibration. BNN shows slight improvements over MC Dropout but does not reach the performance of Deep Ensembles.

| Metric | MC Dropout | BNN | Deep Ensemble |
|---|---|---|---|
| ECE | $0.536 \pm 0.015$ | $0.525 \pm 0.014$ | $\mathbf{0.512 \pm 0.018}$ |
| Brier Score | $0.294 \pm 0.012$ | $0.283 \pm 0.019$ | $\mathbf{0.265 \pm 0.009}$ |
| NLL | $0.805 \pm 0.020$ | $0.794 \pm 0.054$ | $\mathbf{0.731 \pm 0.023}$ |

## 4.3 Qualitative Uncertainty Patterns.

Figure 2 shows NDVI overlays, mean predictions and ground truth label for three representative fire events. The three fire examples shown in Figure 2 span a range of ground-truth sizes: the events span approximately 125.6 acres (large), 52.3 acres (medium), and 5.2 acres (small), corresponding to 1271, 527, and 53 burned pixels respectively. In larger fires, uncertainty is sharply localized near the fire perimeter, while in smaller or fragmented fires, it appears more diffuse and spatially ambiguous. This shows that uncertainty often gathers around the edges of fires—places where the model has to guess how far the fire might have spread, especially when the burn area is small or broken into patches.

## 4.4 Feature Attribution.

To identify which input features (we consider vegetation features only, since they yield best performance) most influence uncertainty, we use Integrated Gradients (Sundararajan et al., 2017) on a ResNet He et al. (2016) surrogate model trained for 50 epochs to approximate the UTAE model's mean predictions. Integrated Gradient computes the path-integrated gradients of the output with respect to input features, highlighting which input pixels most influence predictions. The surrogate model achieves a high fidelity with $R^2 = 0.81$, indicating strong alignment with the original model and validating its use for interpretability. As shown in

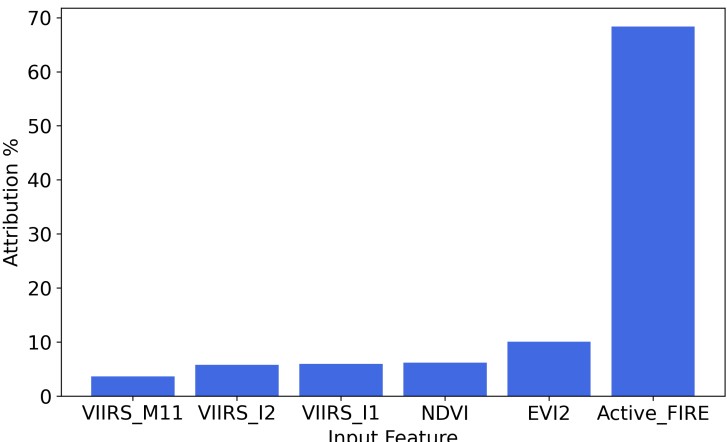

Figure 3: Feature importance scores computed using Integrated Gradients on a CNN surrogate. Active fire presence dominates attribution, followed by vegetation indices (NDVI, EVI2). Thermal bands are less influential in determining predictive confidence.

Figure 3, attribution is dominated by recent fire activity and vegetation indices (NDVI, EVI2), with relatively lower contributions from thermal reflectance bands (VIIRS-M11, I1, I2).This ranking aligns with biophysical intuition: vegetation indices and recent fire activity dominate predictive influence because ambiguous or weak spectral signals in sparsely vegetated or transitional fuel zones make fire spread harder to infer, indirectly contributing to areas where uncertainty later emerges (Archibald et al., 2018).

### 4.5  Centroid-Aligned Boundary Distance as a Proxy for Fireline Uncertainty.

To better understand the spatial structure of false positives in fireline predictions, we introduce a *centroid-aligned boundary distance* metric. For each test instance, we compute the centroid of the ground truth burn mask ($C_f$) and the centroid of the predicted fireline ($C_p$), using the mean prediction from deep ensembled outputs thresholded at 0.95. We then trace a straight line between $C_f$ and $C_p$, and identify the nearest points along this axis where the predicted and ground truth firelines terminate. The distance between these two edge points serves as a localized estimate of spatial prediction error. We formally define the distance metric:

Let $M_{\text{gt}}$ and $M_{\text{pred}}$ be the binary masks for the ground truth and predicted fire regions, respectively. Let $C_f = (x_f, y_f)$ and $C_p = (x_p, y_p)$ denote the centroids of the ground truth region and the false positive region defined as

$$M_{\text{fp}} = M_{\text{pred}} \wedge \neg M_{\text{gt}}.$$

We define the centroid-to-centroid axis as the discrete line segment connecting $C_f$ and $C_p$, denoted by $\mathcal{L}(C_f, C_p)$. Let $\partial M_{\text{gt}}$ and $\partial M_{\text{fp}}$ denote the boundary pixels of the ground truth and false positive regions, respectively. These are computed as:

$$\partial M = \text{dilate}(M) \wedge \neg M.$$

We identify the first boundary pixel $p_{\text{gt}} \in \partial M_{\text{gt}}$ along $\mathcal{L}(C_f, C_p)$ starting from $C_f$, and the first pixel $p_{\text{fp}} \in \partial M_{\text{fp}}$ from the opposite direction. The centroid-oriented boundary distance $d$ is defined as:

$$d = \|p_{\text{gt}} - p_{\text{fp}}\|_2 \cdot s,$$

where $s$ is the pixel resolution. Next, we compare this metric with two established distance measures: Average Surface Distance (ASD) and Hausdorff Distance (HD).

**Average Surface Distance (ASD):** Let $D_{\text{gt}}$ and $D_{\text{fp}}$ be the distance transform maps of the complement regions $\neg M_{\text{gt}}$ and $\neg M_{\text{fp}}$, respectively. For each boundary pixel $p \in \partial M_{\text{gt}}$, we compute its distance to the nearest boundary pixel in $\partial M_{\text{fp}}$ using $D_{\text{fp}}[p]$. Similarly, for each boundary pixel $q \in \partial M_{\text{fp}}$, we compute its

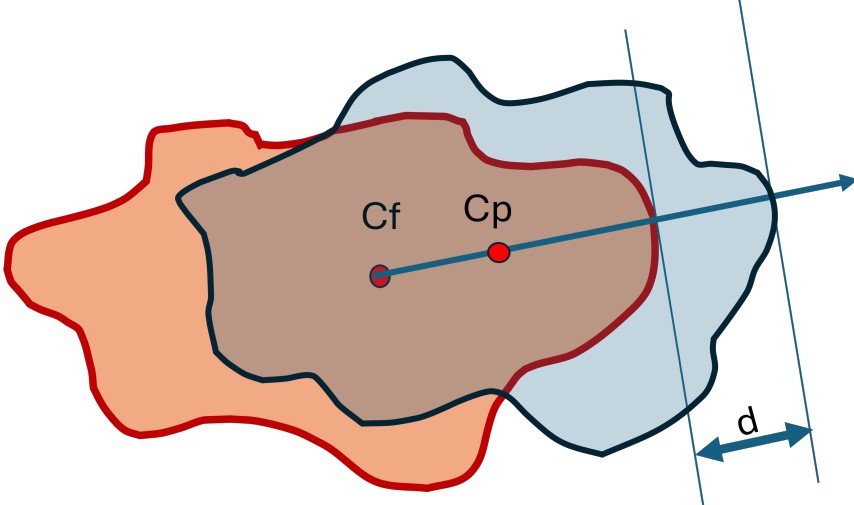

Figure 4: Schematic of boundary distance computation between predicted and ground truth fire masks.

distance to the nearest boundary pixel in $\partial M_{\text{gt}}$ using $D_{\text{gt}}[q]$. The average surface distance $d_{\text{ASD}}$ is defined as:

$$d_{\text{ASD}} = \frac{1}{2} \left( \frac{1}{|\partial M_{\text{gt}}|} \sum_{p \in \partial M_{\text{gt}}} D_{\text{fp}}[p] + \frac{1}{|\partial M_{\text{fp}}|} \sum_{q \in \partial M_{\text{fp}}} D_{\text{gt}}[q] \right) \cdot s.$$

**Hausdorff Distance:** Let $P_{\text{gt}} = \{p : p \in \partial M_{\text{gt}}\}$ and $P_{\text{fp}} = \{q : q \in \partial M_{\text{fp}}\}$ be the sets of boundary pixel coordinates. The directed Hausdorff distances are defined as:

$$h(P_{\text{gt}}, P_{\text{fp}}) = \max_{p \in P_{\text{gt}}} \min_{q \in P_{\text{fp}}} \|p - q\|_2$$

$$h(P_{\text{fp}}, P_{\text{gt}}) = \max_{q \in P_{\text{fp}}} \min_{p \in P_{\text{gt}}} \|q - p\|_2$$

The Hausdorff distance $d_{\text{HD}}$ is defined as:

$$d_{\text{HD}} = \max\{h(P_{\text{gt}}, P_{\text{fp}}), h(P_{\text{fp}}, P_{\text{gt}})\} \cdot s.$$

**Comparative Analysis of Distance Metrics** The three metrics offer complementary perspectives on spatial prediction errors (Table 3). The centroid-oriented boundary distance provides a directionally-informed, single-value summary that captures primary spatial offset efficiently but is limited to one spatial dimension and may miss perpendicular boundary irregularities. The Average Surface Distance (ASD) offers a more comprehensive, symmetric assessment by averaging all boundary distances, providing statistically robust summaries less sensitive to outliers, though it can mask significant local deviations by averaging them with smaller errors. The Hausdorff Distance captures worst-case spatial errors through maximum boundary separation, providing upper bound guarantees valuable for critical applications, but is highly sensitive to noise and isolated mispredictions that may not represent systematic bias. Computationally, the centroid method requires minimal resources, ASD involves distance transforms across boundary pixels, while Hausdorff requires more expensive pairwise calculations. For operational fire management, these three metrics collectively enable rapid spatial offset assessment, balanced characterization of boundary discrepancies, and identification of critical failure cases.

Figure 4 provides a schematic of the centroid-based boundary distance. Aggregating this distance across test samples yields a proxy for the *operational buffer zone*—the typical separation between where the model expects the fire to be and where it actually was. As shown in Table 3, the centroid-based metric peaks consistently around 28–35 meters across all feature sets, indicating relatively small but non-negligible spatial offsets between predicted and observed firelines (approximately 1.5–3.5 Sentinel-2 pixels).

Table 3: Most likely spatial offsets (peak distances) from the KDE histograms for each distance metric and feature group, computed using the deep ensemble model. These values represent the typical separation between predicted and observed wildfire boundaries, providing an empirical estimate of the operational buffer zone.

| Distance Metric | Feature Set | Peak Distance (m) |
|---|---|---|
| Centroid Boundary Distance | Landcover | 28.14 |
| | Topography | 31.26 |
| | Vegetation | 32.19 |
| | Weather | 35.17 |
| | All Features | 33.48 |
| Average Surface Distance (ASD) | Landcover | 46.72 |
| | Topography | 52.89 |
| | Vegetation | 64.15 |
| | Weather | 57.34 |
| | All Features | 55.86 |
| Hausdorff Distance | Landcover | 148.63 |
| | Topography | 153.42 |
| | Vegetation | 165.78 |
| | Weather | 159.11 |
| | All Features | 155.67 |

When considering alternative distance metrics, we observe that Average Surface Distance (ASD) peaks are moderately higher, typically between 47–64 meters, reflecting the fact that ASD accounts for discrepancies across the entire fire boundary rather than a single centroid axis. Hausdorff distances are notably larger ($\approx$148–166 meters), capturing worst-case spatial deviations, such as isolated false positives or missed fire patches. These complementary metrics together characterize prediction errors more comprehensively: centroid distances highlight the main directional bias, ASD summarizes average boundary mismatch, and Hausdorff identifies outlier errors and potential failure cases.

From an operational perspective, these distances are meaningful. A 30–70 meter offset roughly corresponds to a few Sentinel-2 pixels, which is a scale that incident management teams can directly relate to tactical suppression efforts near the active fire edge (Morvan & Dupuy, 2001; Thompson et al., 2016). Larger deviations captured by Hausdorff distances, on the order of 150 meters or more, may indicate localized high-risk regions where the model significantly misaligns with the observed firefront. This information can help prioritize uncertainty-driven buffer zones for contingency planning, resource staging, and safety analysis during wildfire response operations. Given that Sentinel-2 bands used in our models have spatial resolutions of 10–20 meters (Drusch et al., 2012), these metrics offer a practically interpretable and spatially grounded complement to traditional pixel-wise accuracy metrics.

## 5  Discussion

**Limitations**   Even though the dataset includes fire events spanning a wide range of eco-regions across the continental U.S. from forested landscapes to grasslands, it may not fully extrapolate to other geographies such as the Mediterranean or Australia. Testing under such regimes is an important avenue for validating robustness.

Our uncertainty quantification focuses solely on epistemic uncertainty. Aleatoric uncertainty, which can arise from label noise, cloud cover, or unobserved ignitions, is not modeled here but is a natural extension. The PDP approach assumes a logit-normal distribution for per-pixel wildfire probabilities, which may not be the most natural fit for binary segmentation tasks compared to a Bernoulli or Beta distribution. Estimating both mean and variance parameters introduces additional complexity and can lead to unstable or over-dispersed uncertainty estimates, particularly in low-signal or sparse burn regions. Moreover, PDP has been mainly

validated in regression and point forecasting tasks, and its effectiveness for dense spatial wildfire prediction remains less established compared to ensemble-based methods.

The definition of the uncertainty buffer zone using centroid-aligned boundary distance is intuitive and consistent across folds, but it has some caveats. First, this metric does not explicitly follow the actual advancing firefront and instead relies on a centroid-induced direction. In some cases, this axis may point away from the dominant fireline, limiting interpretability for operational decision-making. Second, when multiple disconnected fire foci are present in the ground truth, small peripheral components can disproportionately affect the computed distance, leading to potentially misleading buffer estimates.

Our input feature set is limited to vegetation indices and active fire bands. This constraint is supported by ablation results, which show that including additional inputs like weather, topography, or land cover reduces performance. We speculate this degradation may arise from temporal misalignment, redundancy, or noise. Importantly, the spatial resolution of these feature groups also differs—vegetation indices and land cover layers are high-resolution, while weather variables are typically coarse (e.g., 2.5–10 km grids). For a pixel-level segmentation task, this resolution mismatch introduces a valuable modeling challenge and may explain why weather-derived features failed to help. Exploring fusion strategies that account for spatial scale differences remains a compelling direction for future work.

**Future Work**   Currently, we are extending this work to explore multi-fold and multi-model comparisons of uncertainty dynamics. Another important direction involves improving how high-dimensional multimodal EO inputs are encoded. With data streams including spectral bands, vegetation indices, terrain layers, and fire history, the input space can be both redundant and noisy. We are investigating more effective compression strategies—such as bottlenecked attention, sparse fusion layers, and contrastive pretraining—to ensure that the most informative features drive both prediction and uncertainty, while reducing model complexity and overfitting risk. Testing whether the observed uncertainty patterns generalize to different geographic regions, fuel types, or fire regimes would validate their robustness. Evaluating uncertainty under domain shift (e.g., cross-continent generalization) or in out-of-distribution conditions could also expose failure modes.

## 6   Conclusion

This work presents a systematic analysis of spatial uncertainty in high-resolution, Earth observation-based wildfire forecasting. Using Monte Carlo dropout, deep ensembles, and Bayesian Neural Networks, we demonstrate that uncertainty estimates are not scattered noise but instead form coherent spatial structures aligned with fire perimeters and vegetation gradients. These patterns are quantitatively validated: uncertainty gradients are smooth and consistently form narrow, spatially meaningful bands—typically 20–60 meters wide—around predicted firelines. These findings suggest that uncertainty in EO-based wildfire forecasting is structured, interpretable, and can inform operational decision-making for fire management and risk assessment.

To formalize this observation, we introduce a novel and interpretable centroid-oriented boundary distance metric that quantifies the spatial offset between predicted and ground-truth firelines. This metric reveals a consistent uncertainty buffer zone and offers a practical proxy for operational planning. Through feature attribution, we find that vegetation health and recent fire activity are the strongest drivers of predictive confidence, reinforcing the spatial and temporal grounding of model uncertainty. Additionally, we observe that uncertainty zones scale modestly with fire size, suggesting that predictive uncertainty reflects localized ambiguity rather than arbitrary noise.

Our analyses span multiple architectures and model instances, increasing confidence that the observed uncertainty behaviors are not artifacts of a specific model but reflect generalizable patterns tied to fire morphology and feature dynamics. Overall, these results highlight that spatial uncertainty carries interpretable and actionable structure. Rather than being discarded as noise, it can be embraced as a signal—indicating where model hesitation, boundary ambiguity, or further scrutiny may be warranted. As EO-based systems become increasingly operational, such structured uncertainty maps may support more robust, risk-aware wildfire response and decision-making.

**Broader Impact Statement**

Wildfires represent a growing global threat, exacerbated by climate change, land-use patterns, and increased human activity in fire-prone areas. This work contributes to the development of transparent and trustworthy AI models for high-resolution wildfire forecasting by introducing methods to quantify and spatially interpret predictive uncertainty. Our goal is to improve decision-making for fire managers, emergency responders, and policymakers through spatially calibrated uncertainty estimates that identify where model predictions may be unreliable.

By revealing areas of model hesitation—particularly near fire boundaries and in transitional vegetation zones—our approach may help inform triage strategies, resource allocation, and evacuation planning. The incorporation of uncertainty into operational workflows could reduce overreliance on overconfident predictions, thereby enhancing safety and trust.

However, there are potential risks. Misinterpretation of low uncertainty as a guarantee of safety could be dangerous, especially in out-of-distribution regions. Our models are trained on U.S. fire data and rely on vegetation-based inputs, which may not generalize to other ecosystems. Additionally, we focus on epistemic uncertainty, without modeling aleatoric factors such as sensor noise or unobserved ignitions.

To mitigate these limitations, we recommend human-in-the-loop deployment, transparent communication of model assumptions, and expanded testing across geographies. We also encourage future work to involve diverse stakeholders—including frontline responders and indigenous communities—in evaluating the operational value and limitations of spatial uncertainty maps. Ultimately, this work aims to support safer, more robust wildfire management systems that align with public and environmental benefit.

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

# A Appendix

## A.1 UTAE Model Selection Rationale and Procedural Details

We provide detailed implementation information for the UTAE baseline, including architectural, optimization, and data processing details. We also motivate our choice of UTAE over alternative temporal architectures.

**Model Architecture.** We adopt the UTAE architecture(Garnot & Landrieu, 2021), a U-Net variant with a Temporal Attention Encoder that applies simplified multi-head self-attention across the temporal dimension at the bottleneck. These temporal attention weights are up sampled and applied to the skip connections, enabling dynamic selection of temporally relevant features. The model has approximately 1.1M parameters and includes a dropout rate of 0.1 after each attention block to prevent overfitting.

**Motivation for UTAE.** We chose UTAE for its lightweight parameter count, proven effectiveness on spatiotemporal satellite time series segmentation, and its compatibility with variable-length sequences. Although newer transformer-based architectures, such as Swin Transformers (Liu et al., 2021), have been explored for spatiotemporal wildfire modeling (Lahrichi et al., 2025), empirical evidence suggests that they do not outperform UTAE in next-day wildfire spread prediction. On the downside, these larger architectures contain roughly 27M parameters compared to UTAE's lightweight 1M parameters, requiring significantly more computational resources for both pretraining and fine-tuning. They are also prone to overfitting and typically demand much larger datasets to generalize effectively—an unrealistic requirement given the size and variability of current wildfire datasets. For these reasons, and given UTAE's proven reliability and favorable trade-off between performance, computational efficiency, and robustness, we adopt UTAE as our primary model. Our empirical findings also show that UTAE outperforms ConvLSTM and standard U-Net by a margin of up to 3.9 AP points on this dataset.

**Input Configuration.** Each input sequence consists of 5 days of observations, each containing:

- **Vegetation**: VIIRS reflectance bands (I1, I2, M11), NDVI, EVI2

- **Weather and Forecasts**: Precipitation, temperature (min, max), wind (speed, direction), specific humidity, PDSI, ERC, GFS forecasts of the same

- **Topography**: Slope, aspect, elevation

- **Land cover**: One-hot encoded MODIS IGBP class

- **Fire masks**: Timestamped detection map and binary mask

- **Day-of-year**: Integer mapped to temporal embedding

**Preprocessing** All input features are resampled to a spatial resolution of $375\,\mathrm{m}$ and a temporal resolution of 24 hours. Numerical features are standardized to zero mean and unit variance, excluding angular features (sine-transformed) and categorical/binary maps. Missing values are replaced with zero. We apply the following augmentations:

- Random crop to $128 \times 128$ pixels, with oversampling based on fire presence

- Horizontal and vertical flips, 90° rotations

- Angle-aware adjustment for wind direction and aspect post-rotation

At test time, we apply center cropping to a size divisible by 32 to meet U-Net alignment constraints.

**Optimization**   We use the AdamW optimizer Loshchilov & Hutter (2019) with parameters $\beta_1 = 0.9$, $\beta_2 = 0.999$, learning rate $= 0.01$, and weight decay $\lambda = 0.01$. Dropout (0.1) is applied at each temporal attention block. We use a weighted binary cross-entropy loss for loss function. The model is trained for 10,000 steps. Batch size is 32 during training and 1 during testing.

Table 4: Bayesian U-TAE architecture and training configuration. The BNN version is designed with a comparable parameter count to the deterministic UTAE to ensure a fair comparison of uncertainty quantification methods.

| Category | Component | Details |
|---|---|---|
| Architecture | Model | Bayesian U-TAE (Bayesian neural network version of U-TAE) using variational inference to model weight uncertainty |
| | Input Shape | [Batch, 5, 7, Height, Width] |
| | Output Shape | [Batch, 1, Height, Width] |
| | Bayesian Layers | BayesianConv2d layers with learned mean and variance parameters (prior std = 1.0) |
| | Encoder | 3 blocks, widths=[32, 64, 64]; BatchNorm, ReLU, max pooling in first 2 blocks |
| | Decoder | 3 blocks, widths=[32, 64, 64], bilinear upsampling with skip connections |
| | Output Layers | 1 BayesianConv2d (32 channels) + final 1x1 BayesianConv2d |
| Training Regime | Optimization | Adam optimizer, LR=1e-4, weight decay=1e-5, KL weight=1e-4 (annealed over 20 epochs), gradient clipping norm=1.0,early stopping patience=15 |
| | Epochs | 50 |
| | Loss | ELBO = Binary Cross-Entropy + KL Weight $\times$ KL Divergence |

