# OpenReview forum: "Spatial Uncertainty in Wildfire Forecasting Using Multi-Modal Earth Observation"
_TMLR — Rejected by TMLR_

### Review · Reviewer_8bH1 · 2025-07-09

**Summary Of Contributions:**

This paper investigates uncertainty quantification in wildfire forecasting. Specifically, their method uses a UTAE architecture (a deep network architecture designed for satellite image-time series) to process samples from the WildfireSpreadTS dataset. To quantify the uncertainty of the model's predictions, they perform 20 forward passes per sample _with_ dropout enabled, computing per-pixel mean and variance. They also calibrate logits by tuning a temperature parameter on the validation set. They experiment with four feature groups as model inputs, in addition to active fire inputs that are always present: vegetation, weather, land cover, and topography. They find that vegetation+active fire is the best input. They find that quantifying uncertain with an ensemble of 5 independently trained models is better than quantifying uncertainty using dropout, but the ensemble is costlier to train. Using integrated gradients, they find that the active fires input is the most informative for prediction, which is unsurprising.

Next they introduce a distance metric to help understand the spatial structure of false positives. The metric first computes the centroid of the ground-truth burn mask and the centroid of the fire prediction. The metric then traces a straight line between these two centroids. The authors say this metric estimates the "typical separation between where the model expects the fire to be and where it actually was."

**Audience:**

Yes

**Broader Impact Concerns:**

No concerns

**Claims And Evidence:**

Yes

**Requested Changes:**

To secure my recommendation for acceptance please:
- detail the UTAE training procedure
- compute other distance metrics and compare them to the proposed distance metric
- experiment with other SoTA uncertainty quantification methods and compare them to dropout and ensembling

**Strengths And Weaknesses:**

Strengths:
- The paper is well written and easy to follow
- The problem is well motivated
- A sensible dataset is chosen (WildfireSpreadTS)
- Sensible performance metrics are chosen (e.g., for uncertainty: NLL, Brier Score, and ECE)
- The proposed distance metric is intuitive

Weaknesses:
- The training procedure, e.g., hyper-parameters, is not detailed in the paper (and I do not see a supplement)
- I'm unsure if this paper produces insights that can generalize to other ML applications
- There is no mention of seemingly similar distance metrics like Average Surface Distance (ASD) or Hausdorff distance
- Combinations of multiple input features (e.g., weather _and_ vegetation) are not evaluated

---

### Review · Reviewer_tH7N · 2025-07-17

**Summary Of Contributions:**

This paper presents a short analysis related to the uncertainty of fire growth forecasting models. Results are based on the WildfireSpreadTS dataset using the UTAE model (with some results using a ConvLSTM). Uncertainty is estimated via Monte Carlo Dropout. In addition, the paper proposes a “centroid-aligned boundary distance” that aims at capturing the expected distance between the predicted and the real firefronts, which is intended as an operationally useful metric.

The results show that:
- Vegetation-related inputs, along with the active fire ones, are the most important for the task.
- MC Dropout performs significantly worse than an ensemble of independently trained models when it comes to uncertainty estimation.
- The entropy maps produced by MC Dropout are smoother than noise.
- The “centroid-aligned boundary distance” tends to be of the order of tens of meters.

**Audience:**

Yes

**Claims And Evidence:**

No

**Requested Changes:**

1. I would strongly suggest to revise the message of the paper in order to ensure that all the conclusions are supported by the experimental results, as per weakness 1.
2. I would also suggest to define more clearly the “centroid-aligned boundary distance”. In addition, I think the potential weaknesses raised in the second point should be addressed and, potentially, additional possible metrics suggested that address these weaknesses.
3. I would suggest as well to double check the entropy calculations, which seem to behave in an unexpected way in Fig 2.

**Strengths And Weaknesses:**

This paper tackles an important problem, and does so with the ambition of providing information that could be operationally useful for firefighting, via the proposed “centroid-aligned boundary distance”, which is intended to capture the expected distance between the predicted firefront and the real one.

I do have some reservations about the paper:

1. I would argue that some of the claims in the paper are not supported by the results:
- In the paragraph “Benchmarking Calibration for Trustworthy Spatial Uncertainty”, it is claimed that “This comparison strengthens our confidence that the uncertainty maps derived from MC Dropout are probabilistically reliable, providing a sound foundation for the spatial analyses that follow”. This is based on Table 2, which simply shows that the ensemble performs better than MC Dropout in all metrics. I am not able to see how the results support that conclusion.
- In “Feature Attribution”, the paragraph ends with “This ranking aligns with biophysical intuition: model uncertainty peaks in sparsely vegetated or transitional fuel zones, where spectral signals are ambiguous or weak, leading to higher predictive hesitation”, which seems to be mixing feature importance with uncertainty.
- In “Spatial Coherence of Uncertainty”, where the smoothness of the estimated uncertainty maps is compared to a shuffled map and a fully random one. It is not surprising that the predicted map is smoother than the radomized ones, since it stems from a convolutional model with a strong spatial autocorrelation inductive bias. The conclusion that this means that the uncertainly is “aligned with meaningful structures—particularly the firefront—” cannot be extracted from these results.

2. I would think that the main contribution of the paper is the “centroid-aligned boundary distance”.
- However, this metric is only described textually, with no mathematical expression. This leaves some ambiguity. For instance, is $d$ the same if the predicted firefront is beyond the real one or behind? This is unclear to me, since these two cases are very different from an operational perspective.
- It is also unclear how the approach deals when multiple foci are present. In the examples shown (particularly in the first one), the GT seems to contain several disconnected components, and the small ones may have a very large impact on the metric, affecting its usefulness.
- The metric seems to ignore the actual firefront (I noticed that sometimes the word fireline is used, but I think this usually refers to firebreaks), since the direction induced by the centroids is rather arbitrary and may actually point to the rear of the fire, making again its usefulness less clear.
- There is no experiment designed to measure the potential usefulness of the approach.
3. I have a couple minor comments:
- In Fig 2, it is a bit surprising that the entropy is highest where the model confidently predicts absence of fire. I could imagine that there is some issue with the computation of the entropy here.
- In the paragraph “Qualitative Uncertainty Patterns”, it says that Fig 2 displays three events, when there are only two.

---

### Review · Reviewer_coXq · 2025-07-23

**Summary Of Contributions:**

This paper aims to investigate: how can we predict wildfire from remote sensing data **while estimating uncertainties**? To achieve this, they leverage Monte Carlo Dropout - this allows uncertainty estimates to be made in addition to predictions of wildfire occurrence.

A UTAE model is trained on the WildFireTS dataset, which includes NDVI, EVI, VIIRS, topography, weather and active wildfire. Monte Carlo dropout is used to estimate model uncertainty; the paper then conducts quantitative and qualitative analyses to validate the uncertainty maps.

**Audience:**

Yes

**Broader Impact Concerns:**

I feel the broader impact statement appropriately addresses the ethical implications of the work.

**Claims And Evidence:**

No

**Requested Changes:**

As discussed above, the most important change I have is that the authors spend more time convincing readers that the uncertainty maps are meaningful and useful. The quantitative scores seem low, and the maps in Figure 2 seem to simply reflect the mean prediction, not the model’s correctness. I appreciate the authors attempted to demonstrate this in Figure 3b but random noise seems like a low baseline to beat (an uncertainty map can have much more structure than random noise but still not be useful).

As it is, I am not convinced that the evidence "confirm[s] that these uncertainty estimates are probabilistically well-calibrated across multiple folds".

It’s possible I am misinterpreting the maps and values, in which case it may be helpful to clarify those sections.

**Strengths And Weaknesses:**

The primary contributions of the paper are (i) demonstrating the applicability of uncertainty estimation in this domain, and (ii) further analysis of the uncertainty maps.  This is a well motivated study - uncertainty estimation for wildfire estimation is an important problem. The paper is detailed and easy to read and follow.

My main concern is to do with the quality of these uncertainty estimates; both the quantitative results (Table 2) and the qualitative results (Figure 2) do not make me confident that the models are able to estimate their uncertainty, so I am not confident in the downstream analysis.

### Feature Group Ablations
- [minor] What was the motivation for not training models on all the input modalities, or combinations of the modalities (as is done in the WildFireTS paper) in Table 1’s ablations?
- The paper’s limitation section states “This constraint is supported by ablation results, which show that including additional inputs like weather, topography, or land cover reduces performance” but the ablations do not clearly show this effect to me. It would be helpful to have a “vegetation + topography + active wildfire”, etc. to support this conclusion.

### Benchmarking Calibration for Trustworthy Spatial Uncertainty
- [major] I am confused by the conclusions in this section. Why does achieving better results for all metrics using deep ensembling give you confidence in the Monte Carlo dropout approach? It is not clear to me why Table 2 demonstrates that the Monte Carlo dropout results are probabilistically reliable. A Brier score of 0.25 does not seem (intuitively) very reliable (since it implies a MAE of forecasts of >0.5). Since this motivates all the following analyses, clarifying or better justifying this seems critical.
- [major] Since ensembling does better why is this not the method used for all following analyses?

### Qualitative Uncertainty Patterns.
- [major] In the qualitative analysis (Figure 2), it looks like the entropy very closely overlaps with the mean prediction; it doesn’t seem like the model has a higher entropy where it is wrong and a lower entropy when it is correct. My intuition is that ideally we would have low entropy where the model is correct, and high entropy where the model is wrong; is this the correct understanding?
- [minor] Could the active fire maps be added to Figure 2?
- [nit] Figure 2 and “Qualitative Uncertainty Patterns.” describe three fire events, but there are only two rows (with 2 events).

### Feature Attribution
- [minor] I would appreciate more details about training the ResNet. In particular, what are the inputs for this ResNet? It seems from Figure 3a that topography is not included; why is this the case?

---

### Review · Reviewer_ktR2 · 2025-07-23

**Summary Of Contributions:**

This paper proposes to explore uncertainty quantification for wildfire forecasting. Their forecasting setup is based on image time series with various modalities, notably including binary masks of the fire at previous time steps and Sentinel-2 reflectance information relevant to the detection of vegetation. The author's strategy leverages UTAE, a deterministic satellite image time series processing model previously introduced for crop type segmentation. They propose to use Monte-Carlo dropout and deep ensembling to make the predictions probabilistic, and assess the quality of the obtained results through various uncertainty-aware metrics, with some attention to the spatial structure of the uncertainties.

**Audience:**

Yes

**Claims And Evidence:**

No

**Requested Changes:**

- In the introduction the authors make a detailed account of damages caused by wildfire throughout the world. While it is important to give some context, I feel like it is unnecessary to go into so much detail for a short machine learning article. For exemple, instead of "in January 2025, Southern California experienced [...] marking them among the costliest natural disasters in U.S. history", I would suggest just saying something like "in January 2025, Southern California experienced wildfire events that have been marked among the costliest natural disasters in U.S. history". Besides, the authors should keep in mind that scientific articles are usually more valuable references than other kinds of references.

- Two different references are given for the UTAE paper but it seems that none of them is correct: one has a wrong title and the other is "arxiv 2022" instead of "ICCV 2021". I suggest checking all the references in detail because there might be other similar mistakes there.

- Regarding weakness (2), I think it is questionable to base the study on the relatively old UTAE paper considering how rich and active the satellite image time series processing field is (see e.g. [1]). I think that having a more extensive benchmark including more recent models would be very valuable. If the authors are not able to do this, they should at least comment on why they chose UTAE.

- The same suggestion can be made for the method used to make the backbone models uncertainty-aware: dropout is a simple baseline but usually not the best one, see e.g. [2] for alternatives.

- Concerning weakness (3), for their main prediction task, the authors explain that "for each fire event, we extract 5-day sequences as input and predict the binary burn mask at the 6th day". For a non-specialist, this sounds like this task is relatively easy, in the sense that the persistence baseline (i.e. predicting that the mask for the 6th day is the same one as on the 5th) should give you decent estimates already. So I suggest the authors do two things: 1) Show some instances of the task, with the 5 input masks and the corresponding groundtruth on the next day, in order to give readers an intuitive sense of the difficulty of the task. 2) Display the performance (e.g. average precision) of the persistence baseline in order to know how well your methods perform compared to this trivial approach.

- As pointed out by other reviewers, the entropy scale in figure 2 seems counterintuitive, and the figure's legend as well as the main text talk about 3 wildfire events while only 2 are represented.

- The authors show that deep ensembles perform better than MC dropout and conclude from this that they have confidence in the MC dropout calibration, which is quite confusing. By the way, are the members of the deep ensemble deterministic or do they all use MC dropout?

- As pointed out by other reviewers, the arguments for showing that uncertainty estimates are spatially coherent are not convincing. Probably plotting some MC samples might be more convincing. It would be interesting to show that individual MC samples can be interpreted as plausible scenarios for the evolution of a wildfire.

Regarding weakness (4), the authors should write a formal definition of their new metric, in addition to the textual explanation. The schematic of figure 4(a) is quite confusing since it seems to show that the line between centroids is bounded only on one side: is this intentional?

**Minor comments**

- Putting best results in bold in all tables would be preferable.

- The authors only draw 20 Monte Carlo samples from the model with dropout, but usually the performance improves when drawing more samples, so I would suggest trying it.

- The authors should provide some explanation of what integrated gradients consist in, as well as at least one reference

- The acronym "KDE" is undefined.

[1] Miller, Lynn, Charlotte Pelletier, and Geoffrey I. Webb. "Deep learning for satellite image time-series analysis: A review." IEEE Geoscience and Remote Sensing Magazine 12.3 (2024): 81-124.)

[2] Haynes, Katherine, et al. "Creating and evaluating uncertainty estimates with neural networks for environmental-science applications." Artificial Intelligence for the Earth Systems 2.2 (2023): 220061.

**Strengths And Weaknesses:**

**Strengths**

(1) The authors are seemingly the first ones to consider uncertainty quantification for wildfire forecasting models, which is an interesting and well-justified problem.

(2) The public sharing of the code, which is rather cleanly written and enables to reproduce the manuscript's experiments, is appreciated.

**Weaknesses**

(1) The paper critically lacks implementation details, such as the dropout proportion, optimizer algorithm and its parameters, etc. Likewise, some explanations are missing for existing methods used in the paper (e.g. integrated gradients and post-hoc temperature
scaling) or reasons why the authors chose to use UTAE among other similar architectures (many of which are more recent).

(2) The authors express the ambition to establish a benchmark for uncertainty quantification in wildfire forecasting, but the current benchmark is rather weak in terms of both quantity and quality of the implemented approaches.

(3) The lack of previous works addressing the same problem means that the authors should be more careful about making sure (and showing) that the obtained performance are non-trivial. Specifically, they do not give the readers visual or statistical insight on how difficult the task actually is. Since the presented visual and quantitative results are rather unimpressive, I have concerns that they might not actually beat a trivial baseline such as persistence.

(4) The "centroid-aligned boundary distance", which seems to be the main novelty of the manuscript, is described in an informal way that does not enable sufficient clarity.

---

### Decision · Action_Editor_jCPL · 2025-09-30

**Recommendation:** Reject

**Additional Comments:**

The work cannot be accepted at this time, but is welcome to re-submission with substantive changes to respect and incorporate the feedback from this round of review at TMLR. The action editor encourages the authors to consider the scope of the claims and the provided evidence alongside the messaging for the intended audience for the empirical results and code. The reviewers are representative of the audience as researchers and practitioners in machine learning for remote sensing, and so revision in these dimensions may help the reception of the work as a paper and more more generally.

Note this work exceptionally received four reviews for certainty given diversity of opinion. With all four reviewers and the action editor in agreement, given expertise spanning machine learning and remote sensing, the decision reflects all of the perspectives shared in this review process.

**Audience:**

No

**Audience Explanation:**

Reviewers were split on the audience criterion with two votes yes and two votes no. The action editor has closely examined the work, the reviews, and discussion. While machine learning for remote sensing is a valid, established, and important topic that does not satisfy the audience criterion in itself: work must inform the community for this topic in research and applications. Due to the limited scope of the methodology and experiments the audience criterion is not satisfied. To satisfy this criterion more careful explanation and justification of the metrics is needed, more qualitative results than a single prediction per fire for three sizes of fire are needed, and correct reference (note issue with UTAE reference for example) and comparison to existing work in the experiments. Machine learning for remote sensing now has multiple and open options for modeling, not only the UTAE but pre-trained models like AnySat/MMEarth/Galileo/Tessera/etc. and so further experiment is a fair requirement to ensure audience.

**Claims And Evidence:**

No

**Claims Explanation:**

The claims exceed the evidence in multiple respects and by a sufficient margin to consider this disconnect as justification for rejection. All reviewers agree that the claims and evidence criterion is not satisfied. In particular:

- "operationally actionable" is a high bar set by this submission for itself: evidence for this would show action that is being taken or has been taken due to the system but this is absent (coXq, tH7N, 8bH1, ktR2)
- proposed metrics make claims about the relationship of their measures and relevant properties of fires, but is confused in its terminology ("firefront" / "fireline" / "firebreak") and does not distinguish active vs. inactive boundaries of the fire in a clear way that connects claims to evidence (tH7N)
- the uncertainty modeled not sufficiently characterized quantitatively or qualitatively (tH7N, ktR2) in that the single Fig. 2 does not show patterns across multiple inputs of multiple events
- as a more minor but nevertheless relevant point, certain claims are imprecise and could be made more exact, such as the "analyses span multiple architectures" when the number is more precisely two (tH7N, ktR2)

Note that the paper is evaluated as technically correct, but this does not imply that it's claims and evidence are in agreement, and in this case they are not. The action editor agrees with the individual and joint assessment of the reviewers and their discussion. Certain reviewers expressed the work is somewhat borderline in certain aspects, in that the topic is relevant and studying uncertainty is valid, but overall the work exceeds a fair interpretation of the claims and evidence.

**Resubmission Of Major Revision:**

The authors may consider submitting a major revision at a later time.